# Evolutionary Comparisons of Chelonid Alphaherpesvirus 5 (ChHV5) Genomes from Fibropapillomatosis-Afflicted Green (*Chelonia mydas*), Olive Ridley (*Lepidochelys olivacea*) and Kemp’s Ridley (*Lepidochelys kempii*) Sea Turtles

**DOI:** 10.3390/ani11092489

**Published:** 2021-08-25

**Authors:** Liam Whitmore, Kelsey Yetsko, Jessica A. Farrell, Annie Page-Karjian, Whitney Daniel, Donna J. Shaver, Hilary R. Frandsen, Jennifer Shelby Walker, Whitney Crowder, Caitlin Bovery, Devon Rollinson Ramia, Brooke Burkhalter, Elizabeth Ryan, David J. Duffy

**Affiliations:** 1Whitney Laboratory for Marine Bioscience and Sea Turtle Hospital, University of Florida, St. Augustine, FL 32080, USA; kyets001@fiu.edu (K.Y.); jessicafarrell@whitney.ufl.edu (J.A.F.); devonrenee@whitney.ufl.edu (D.R.R.); bmburkhalter@whitney.ufl.edu (B.B.); duffy@whitney.ufl.edu (D.J.D.); 2Department of Biological Sciences, School of Natural Sciences, University of Limerick, V94 T9PX Limerick, Ireland; Elizabeth.Ryan@ul.ie; 3Department of Biology, University of Florida, Gainesville, FL 32611, USA; 4Harbor Branch Oceanographic Institute, Florida Atlantic University, Fort Pierce, FL 34946, USA; cpagekarjian@fau.edu; 5South Carolina Aquarium, 100 Aquarium Wharf, Charleston, SC 29401, USA; wdaniel@scaquarium.org; 6Division of Sea Turtle Science and Recovery, Padre Island National Seashore, Corpus Christi, TX 78480, USA; donna_shaver@nps.gov (D.J.S.); hilary_frandsen@nps.gov (H.R.F.); Jennifer_Shelby_Walker@nps.gov (J.S.W.); 7Gumbo Limbo Nature Center, Boca Raton, FL 33432, USA; WCrowder@ci.boca-raton.fl.us (W.C.); CBovery@ci.boca-raton.fl.us (C.B.)

**Keywords:** CFPHV, ChHV5, phylogenetics, phylogenomics, viral evolution and diversity, marine turtles, fibropapillomatosis

## Abstract

**Simple Summary:**

Our research aims to unravel uncertainties relating to the genetic and viral causes of the debilitating sea turtle disease fibropapillomatosis, which affects all seven species of sea turtle. This disease is likely caused by an alphaherpesvirus (ChHV5) and an environmental trigger (e.g., pollution). Fibropapillomatosis is characterised by multiple benign tumours which grow on the skin, eyes and internal organs, and is becoming a threat to sea turtle conservation globally. ChHV5 research is crucial to better provide effective management and conservation of turtles from this disease. This study aimed to compare ChHV5 genomes between geographic regions and sea turtle species and observe how this virus has evolved and changed. ChHV5 genomes harboured differences within and between geographic regions (88–2793 single nucleotide polymorphisms (SNPs) per sequenced genome). Multiple ChHV5 genes were also found to be under varying selective pressures. Phylogenomic and phylogenetic analyses revealed grouping of the virus, mostly by geography rather than by species, and found differences in ChHV5 genomes between tumours from the same individual. This study pioneers the phylogenomic approach to ChHV5 research. This study provides the most comprehensive picture to-date of whole-genome inter-species ChHV5 diversity and provides important baseline ChHV5 genomic data for future comparisons.

**Abstract:**

The spreading global sea turtle fibropapillomatosis (FP) epizootic is threatening some of Earth’s ancient reptiles, adding to the plethora of threats faced by these keystone species. Understanding this neoplastic disease and its likely aetiological pathogen, chelonid alphaherpesvirus 5 (ChHV5), is crucial to understand how the disease impacts sea turtle populations and species and the future trajectory of disease incidence. We generated 20 ChHV5 genomes, from three sea turtle species, to better understand the viral variant diversity and gene evolution of this oncogenic virus. We revealed previously underappreciated genetic diversity within this virus (with an average of 2035 single nucleotide polymorphisms (SNPs), 1.54% of the ChHV5 genome) and identified genes under the strongest evolutionary pressure. Furthermore, we investigated the phylogeny of ChHV5 at both genome and gene level, confirming the propensity of the virus to be interspecific, with related variants able to infect multiple sea turtle species. Finally, we revealed unexpected intra-host diversity, with up to 0.15% of the viral genome varying between ChHV5 genomes isolated from different tumours concurrently arising within the same individual. These findings offer important insights into ChHV5 biology and provide genomic resources for this oncogenic virus.

## 1. Introduction

Fibropapillomatosis (FP) is a debilitating neoplastic disease which has been reported in all seven species of sea turtle [1], these species range from vulnerable to critically endangered [2]. The disease has a global spread, but with prevalence in specific populations varying considerably [3,4,5,6,7,8]. First described in the scientific literature in the 1930′s [9], This disease is most prevalent in green turtles (*Chelonia mydas*), which also tend to be the most severely afflicted; however, FP has been documented, to a lesser extent, in all other species [5,7,10,11]. Fibropapillomatosis manifests as multiple tumours that primarily arise from the soft tissues of sea turtles, including: cutaneous, ocular and visceral tumours (fibromas, fibrosarcomas, mixofibromas and mixomas), which can vary in size and distribution [12,13]. These tumours can be severely debilitating; impairing vision, locomotion, feeding, predator evasion and other natural behaviours, and preventing affected turtles from providing their valuable ecosystem services and keystone species functions [1,11,12,14,15].

This disease also afflicts turtles at crucial life-stages; juvenile turtles develop FP following recruitment from the oceanic zone into their neritic foraging areas [16]. Fibropapillomatosis is one of the most significant transmissible diseases known in sea turtles and remains a persistent health concern despite conservation successes and significant growth of some affected populations [17]. The FP epizootic has been identified as one of the five major threats to sea turtles, which has been reflected in the renewed scientific interest in this disease in the last decade [5,7,11,13,18,19,20,21,22,23,24,25,26]. Prevalence statistics reveal the rapid establishment of FP among many sea turtle populations, with reported increases from 13.3 to 42% in Florida, USA (2005–2016), 13.2 to 35.3% in northeastern Brazil (2012–2015), 0% to 33% in Guinea-Bissau (2009–2019) and 0.6% to 35.2% in Texas, USA (2010–2018) [7,17,27,28,29,30,31]. The disease also continues to be reported in previously unaffected populations [32,33,34,35]. Increases in incidence such as these are particularly worrying as turtles are thought to have robust anti-cancer defences given the rarity of other forms of neoplasia [7,36].

Chelonid herpesvirus 5 (ChHV5, an alpha herpesvirus) is FP’s putative aetiological agent based on transmission studies and molecular detection of CHV5 in tumours [19,37]; however, an inability to isolate the virus prevented fulfilment of Koch’s postulates during early foundational studies [6,11,22,38].

Molecular studies have consistently detected ChHV5 presence in turtles with FP; however, FP may also be found in turtles where ChHV5 is not detected [26,39], and ChHV5 is detectable in turtles without FP [40,41]. Recently, the green turtle (*C. mydas*) papillomavirus, CmPV1, was detected in 47% of FP tumours analysed from green turtles in Australia, despite earlier conventional PCR-based approaches and whole genome sequencing failing to detect CmPV1 in FP tumours [25,26,42]. This suggests that other oncogenic viruses may contribute to the development of FP. Transmission of ChHV5 likely occurs through direct contact, shedding of virus from viral laden tumours into the environment and through vectors such as marine leeches [25,43,44]. No transmission data currently exists regarding PV1.

Multiple researchers have linked the occurrence of FP with various forms of anthropogenic habitat degradation [45,46,47,48], leading to the current hypothesis that the disease is caused by viral infection in conjunction with environmental co-factors. Such previous data indicate a latent state of this virus, which may recrudesce in times of immunological stress, enabling ChHV5 loads to pass an oncogenic threshold [5,7,20,24,25]; however, specific co-factors and their role in tumourigenesis have yet to be identified. FP has occurred in isolated regions globally within a relatively short timeframe, with differing geographic variants making it unlikely that recent virulence mutations in the virus independently evolved to drive these outbreaks [14]. “It is far more likely that changes in the environment or ecological factors that affect virus transmission or disease expression explain the recent upsurge in disease prevalence almost simultaneously” and “these disease outbreaks are likely induced by environmental factors rather than the virus transmitting to new populations or undergoing mutational adaptation” [14].

Phylogenetics has been used to investigate ChHV5 transmission dynamics [14,16] and to study the evolution of ChHV5, and phylogenetics has identified a number of regional variants [4,5,14,16,35,44,48,49,50,51]. These studies showed that the global distribution of CHV5 in sea turtle populations predates the awareness of an FP epizootic in the 1980s and 1990s, suggesting that co-factors contributed to disease emergence [14]. While several ChHV5 variants have been identified, no viral variant has ever been associated with disease severity or outcome. Studies have found that, at a local scale, sympatric species of sea turtle can share variants of ChHV5, indicating a strong geographic influence on viral phylogeny [4,14,23,48].

Nevertheless, further classification of variants based on the entire ChHV5 genome may enhance our understanding of ChHV5 evolution, the spread of the virus and detection and interpretation of emerging mutations [39]. Furthermore, ChHV5 genomic studies may help explain slight differences in disease manifestation in turtles from different regions (e.g., high prevalence of oral tumours in Hawaiian green sea turtles).

To date, global phylogeography of ChHV5 has been explored somewhat. Herbst et al. (2004) [4] identified two major global clades of ChHV5, each with Atlantic and Pacific strains. Further, Patrício et al. (2012) [48] proposed four major clades: eastern Pacific, mid-west Pacific, western Atlantic/eastern Caribbean and Atlantic. Greenblatt et al. (2005) [52] also identified a ChHV5 variant from Puerto Rico, which, at the time, did not cluster with any other known ChHV5 variant, but has since been clustered with Gulf of Guinea variants [48]. On a more local scale, distinct variants have been identified in some locations. Florida has four known variants of ChHV5 (known as variants A, B, C and D (variant D only from *C. caretta*)), as well as Hawaiian variants and, more recently, Australian variants identified in Queensland [5,14].

The individual genes frequently used for ChHV5 phylogenetics include UL18, UL30, glycoprotein B (gB) and F-sial [8,12,48]. Conventional PCR coupled with Sanger sequencing of individual gene fragments has been the predominant technique to date for ChHV5 phylogenetic analysis.

Relying on short individual gene fragments has yielded significant results but is somewhat restrictive and can lead to a limited picture of the true genetic and phylogenetic diversity amongst ChHV5 variants globally [39].

The first study to construct a large multi-gene sequence of ChHV5 was carried out by Herbst et al. (2004) [4], who configured a partial genome 43,843 bp in length (genes UL9-30). Currently, the most complete ChHV5 reference genome constructed is 132,233 bp long, primarily only lacking repeat regions [53]. Morrison et al. (2018) [39] established large multi-gene sequences to compare ChHV5 gene diversity from eight tumour samples, using short-read Next Generation Sequencing (NGS) of long-read PCR products of 72,828 bp in length (roughly 55% of ChHV5′s current known genome size) aligned to the ChHV5 reference genome [53]. Morrision et al. (2018) [39] also used a smaller subset of genes (Amplicons IV, V, UL30 and gB) of 6280 bp in length for phylogenetic analysis. Increasingly, Next Generation Sequencing (NGS) approaches are being utilised more widely to study ChHV5 [6,7,20,24,25,39], as this powerful analysis tool can provide comprehensive genomic data of study organisms.

Only one whole-genome phylogenomic study of a chelonian herpesvirus has been conducted to date, by Origgi et al. (2015) [54], who used NGS methods to construct and observe the phylogeny of testudinid herpesvirus 3 (TeHV3), a close relative to ChHV5.

To advance our understanding of genome-level ChHV5 diversity across sea turtle species within the eastern USA, we applied NGS-based approaches to 20 novel FP tumour samples collected from three species of sea turtle. Using these whole-genome data we conducted ChHV5 phylogenomics and investigated ChHV5′s genomic diversity and evolution.

## 2. Methods and Materials

### 2.1. Tissue Sampling

FP tumour samples were obtained from sea turtles that stranded in Florida, Texas, South Carolina and Massachusetts, USA. This research was conducted under sea turtle permit numbers MTP-21-236 and MTP-21-139 from the Florida Fish and Wildlife Conservation Commission and South Carolina Department of Natural Resources (MTP-2019-0005), U.S. Fish and Wildlife Service Endangered Species Permit (TE840727-3) and Texas Parks and Wildlife Department Scientific Permit (SPR-0190-122), and with ethical approval from the Institutional Animal Care and Use Committees (IACUC) at the University of Florida, Florida Atlantic University and National Park Service (201909289). Samples were obtained during tumour removal surgery, or necropsy from rehabilitating or stranded (deceased) green (*C. mydas*), Kemp’s ridley (*L. kempii*) and olive ridley (*L. olivacea*) sea turtles (Appendix A). Unfortunately, loggerhead (*Caretta caretta*) tumour samples were not available at the time of sequencing for inclusion in this study. Samples were obtained as part of separate studies to investigate host and viral dynamics of ChHV5 in tumour samples; for full sampling details please see the respective papers [7,23,25,55]. Samples (from both internal and external tumours) were stored in RNA-later (Qiagen, Hilden, Germany) at −80 °C, or dry at −80 °C, until extraction. Samples were stored between <1 day and 27 months prior to DNA isolation.

### 2.2. DNA Isolation, Library Preparation and Sequencing from Tissue and eDNA Samples

Sequencing of samples was conducted as part of separate studies to investigate host and viral dynamics of ChHV5 in tumour samples, 13 green sea turtle samples [7,25], 6 Kemp’s ridley samples [23] and an olive ridley sample [55]. Sampling and sequencing details are provided in the respective papers [7,23,25,55]. Briefly, DNA was extracted using a DNeasy Blood & Tissue Kit (Qiagen, Cat No. 69504), and all samples were sequenced in an untargeted manner (whole genome sequence of host and viral genes) on an Illumina HiSeq300 (1 sample) or NovaSeq6000 (15 samples), with the exception of four of the Kemp’s ridley samples, for which viral enrichment was performed using an Illumina HiSeq300 platform (4 samples) [23]. Viral enrichment was conducted to assess its potential as a more cost-effective approach to ChHV5 whole genome sequencing.

### 2.3. Quality Control and Read Trimming

All bioinformatic processing was conducted on the Galaxy platform (https://usegalaxy.eu/, accessed on 5 August 2020). The software FastQC—https://www.bioinformatics.babraham.ac.uk/projects/fastqc/, accessed on 5 August 2020—was used to assess data quality. Reads were then trimmed with trim_galore (The Babraham Institute, version 0.5.0, https://www.bioinformatics.babraham.ac.uk/projects/trim_galore, accessed on 5 August 2020) to remove adapter ends with a Phred quality score <20, remove adaptor sequences and remove sequences fewer than 20 bp. For any samples that contained overrepresented sequences according to FastQC, the trimmomatic tool (version 0.36) was then used to remove these sequences from reads and any sequences <25 bp after trimming. The number of raw reads per sample and reads remaining after trimming can be found in Appendix A.

### 2.4. Read Alignment

Reads from all samples were first aligned (paired-end) to the ChHV5 genome (GenBank accession number: HQ878327.2) using Bowtie2 (version 2.3.5.1) on Galaxy, “analysis mode” and “SAM/BAM options” were set to default. The reference ChHV5 genome was derived from a Pacific (Hawai’i) green sea turtle afflicted with FP. The overall alignment rate to the ChHV5 genome was low, with most reads aligning to the green turtle genome (NCBI GenBank Accession numbers: GCA_000344595.1 and GCA_015237465.1), as expected (Appendix A).

### 2.5. Consensus Sequence Generation

Once aligned, count tables (htseq-count, version 0.9.1) for each Bowtie2 alignment were produced on Galaxy, also using the ChHV5 gene annotation file. To determine if each gene had sufficient reads for consensus sequence generation, transcript per million (TPM) values for each gene were calculated manually in Excel.

The ChHV5 Bowtie2 alignments (BAM files) were used as input for Ococo (version 0.1.2.6) to generate consensus sequences for each sample [56]. The reference ChHV5 genome was also selected as the ‘backbone’ of the new consensus sequences. The strategy for building the consensus sequences was performed on a majority basis, with Ococo inferring single nucleotide polymorphisms (SNP’s) on a majority basis, and then a new consensus sequence for downstream analysis based on aligned reads was constructed. Consensus sequences for each ChHV5 genome are provided in Appendix A and have been deposited into the Dryad repository: https://doi.org/10.5061/dryad.wwpzgmsk6, accessed on 5 August 2020).

### 2.6. Nucleotide and Gene Diversity Analysis

The consensus sequences were used to generate nucleotide diversity data and identify positive and reduced selection processes of each ChHV5 gene (Appendix A). Each gene was isolated from each of the 21 genome sequences (including reference, which was used for comparison) using extractseq (version 5.0.0) on Galaxy, inputting gene regions and opting to extract each region to a new sequence.

Next, a purpose-written script was created (deposited in Github: https://github.com/klyetsko/Whitney-SeaTurtle-FP, accessed on 5 August 2020), which first replaced the header for each gene (to the name of the origin sample as well as gene position in genome), then each gene was separated from each consensus sequence into a new file, resulting in 104 files (one file for each of the known 104 ChHV5 genes), with each file containing the sequences for that gene from all 21 samples (one reference sequence and 20 consensus sequences).

The resulting gene text files were then input into DnaSP (version 5) for ChHV5 gene-by-gene analysis to the reference. To note, for the next step, the reference genome sequences were the first sequence in each file, so the DnaSP programme can use that for comparison. Each ChHV5 gene file was opened in DnaSP, selecting “DNA divergence between populations” and “polymorphism/divergence data” to obtain the relevant nucleotide diversity statistics, including number of polymorphic sites, nucleotide diversity and Tajima’s D statistic, which can infer selection pressures under the right demography for each ChHV5 gene (Appendix A).

### 2.7. Phylogenetic/Phylogenomic Analysis

Consensus sequences were input into MEGA X for phylogenetic analysis. For phylogenetics/phylogenomics of generated consensus sequences, whole genomes and relevant genes (ChHV5 UL30, at 2019 and 483 bp) were isolated using Range Extractor (https://www.bioinformatics.org/sms2/range_extract_dna.html, accessed on 5 August 2020) and were compared with known available gene sequences (of the same length and position) from the NCBI database. For phylogenomic analyses, a new alignment build (alignment of inputted sequences (either consensus genomes or isolated genes)) was created, and the entire alignment was then exported in Mega format. Phylogenetic/genomic trees were constructed using the Maximum Likelihood method (Tamura–Nei model).

### 2.8. Patient “Yucca” Whole Genome Phylogenomics

Samples of seven tumours (three kidney tumours and four external tumours (from the left inguinal, right inguinal, tail and right eye)) from patient Yucca (patient ID: 49-2019-Cm, female) were sequenced, and the ChHV5 genome sequence present in each was compared phylogenomically. Yucca had previously been treated for external FP and successfully released tumour free on the 17th October 2018, from the Gumbo Limbo Nature Centre rehabilitation facility (Boca Raton, FL, USA). However, she later re-stranded in northeast Florida with well-developed recurrent FP tumours (9th October 2019) and was cared for at the University of Florida’s Whitney Lab Sea Turtle Hospital until pulmonary and renal tumours were diagnosed by CT scan, and euthanasia was performed for humane purposes (15 October 2019).

## 3. Results

Twenty novel FP tumour samples from 13 sea turtles were utilised for this study (Table 1). Whole genome sequences from 6 Kemp’s ridley samples, 1 olive ridley sample and 13 green sea turtle samples were analysed for ChHV5 aligning reads. ChHV5 genome coverage ranged from 683× to 16,290×x coverage (average of 10,341×) for virally enriched samples and from 7× to 585× coverage (average of 192×) for non-enriched samples (Table 1).

### 3.1. Sequence/Nucleotide Diversity

Firstly, the number of SNPs in the ChHV5 genomes of each sample was compared to the reference genome to measure overall diversity occurring within the virus from each sample (Figure 1A). Most ChHV5 consensus sequences had a high number of SNPs (average number of SNPs, 2035, 1.54% of the ChHV5 genome), compared to the reference genome, suggesting a high level of sequence diversity has arisen in ChHV5 variants (Atlantic vs. Pacific), either through active evolutionary pressure or passive drift. The ChHV5 genome obtained from a green turtle lung tumour (27L1Fdna) had the highest degree of divergence from the reference genome, with 2793 SNPs, or 2.11% of the ChHV5 genome. All samples with >1000 SNPs were derived from three species of sea turtle; Kemp’s ridley, green and olive ridley. The four virally enriched samples used in this study (from Kemp’s ridley turtles) have the lowest number of SNPs, possibly arising from the methodological difference. ChHV5 is predominantly latent in FP tumours (Farrell et al., 2021), and the viral enrichment was based on centrifugation and extraction of DNA from supernatant. Therefore, intracellular latent ChHV5 DNA may have been underrepresented.

The level of SNPs found in ChHV5 genomes obtained from Kemp’s ridley samples was similar to that of the samples taken from green sea turtles (for samples sequenced without viral enrichment). Similarly, the sample obtained from an olive ridley turtle had a high number of SNPs (1434 SNPs), though these were approximately half the number observed in the Kemp’s ridley or green turtle samples (Figure 1A). Interestingly, green turtle-derived ChHV5 genomes from Florida had a higher number of SNPs, when compared to the Hawaiian green turtle-derived reference ChHV5 genome, than either the Kemp’s ridley- or olive ridley-derived ChHV5 genomes. Next, we investigated the diversity and selection of each ChHV5 gene from all consensus sequences.

The Tajima’s D statistic was calculated to determine which individual ChHV5 genes were under the greatest selective pressure. Tajima’s D was calculated by pooling all 20 novel ChHV5 sequences from this study and comparing them with the reference ChHV5 genome (Hawai’i), gene-by-gene. There was a broad range of Tajima’s D values across the individual genes within the pooled ChHV5 genomes (Figure 1B). A small fraction of genes had a value at or close to zero, indicating they were under neutral evolutionary pressure. The majority of ChHV5 genes had changes with a Tajima’s D which deviated from zero (non-neutral), which indicates either demography effects or selective pressures. In the absence of demography effects, genes with negative Tajima’s D values (approximately 55% of the ChHV5 genes) are thought to be under positive selection (represents excessive low-frequency SNPs). Actively conserved sequences and genes with positive Tajima’s D values (approximately 45% of the ChHV5 genes) indicate balancing selection (actively maintained allele diversity) [39,57,58].

Interestingly, the two genes with the highest Tajima’s D values were both tegument proteins (F-UL37 and F-UL36), suggesting that diversity in tegument proteins is maintained and beneficial to ChHV5 survival and propagation (Table 2). The ChHV5 gene under the strongest positive selection was F-UL10 (Glycoprotein M) (Table 2), suggesting that the sequence of this gene is too critical to ChHV5 function to allow large amounts of sequence diversity to evolve. F-UL41, otherwise known as Virion Host Shutoff (VHS) protein, which is also of interest, is known to play a role in evading host innate immunity in other organisms and is highly conserved between alphaherpesviruses [59,60,61].

These results corroborate those reported by Morrison et al. (2018) (across nine tumour samples, including the reference genome (6 Hawai’i/3 Florida)), who examined approximately 63% of ChHV5′s genes. Both sets of analyses demonstrated a wide range of Tajima’s D values across individual genes (Appendix A). However, the specific Tajima’s D value and direction (positive or negative) of each gene varied widely between the two studies (Appendix A). This may be due to the predominance of Hawaiian ChHV5 samples in the Morrison et al. study, the predominance of eastern US samples and the inclusion of Kemp’s ridley and olive ridley samples in the current study.

At the whole genome level, ChHV5 genomes from both green and Kemp’s ridley turtles had positive Tajima’s D values when compared to the Hawaiian ChHV5 reference genome (green turtle derived): 0.077 and 0.546 Tajima’s D, respectively. Kemp’s ridley samples had a higher rate of balancing selection, as indicated by a higher positive Tajima’s D value (Figure 1C). While more individual genes (55%) had a negative Tajima’s D value (Figure 1B), the genome-wide comparison includes non-coding genomic regions, which may be responsible for the skew towards positive values (Figure 1C).

### 3.2. Phylogenomics Reveals Clustering of ChHV5 by Geographic Trends

In order to make whole-genome phylogenomic comparisons, all 20 sequenced samples had genome consensus sequences generated for phylogenomic/genetic analysis against the reference ChHV5 genome. Due to limited available ChHV5 genomic data, aside from our 20 samples [7,23,25,55], the only publicly available ChHV5 genome currently available for comparison is the reference genome [53], which originated from a Hawaiian green turtle (*C. mydas*). Therefore, all consensus sequences were compared to this sole reference genome (Figure 2). To analyse phylogenetic relationships between our 20 consensus genome samples and ChHV5 from other geographic regions, individual gene fragment approaches were used (Figure 3A,B).

At the whole genome level, all 13 novel green turtle-derived ChHV5 genomes (eastern US) generated as part of this study cluster together but form a discrete grouping which is distinct from the green turtle-derived ChHV5 reference genome (Hawai’i) (Figure 2). It is interesting there is not any segregation in clustering between ChHV5 genomes from South Carolina and Florida, suggesting a similar transmission source, despite FP only recently being reported in the Carolinas [22]. The next cluster is represented by ChHV5 from two Kemp’s ridley individuals, also from the east coast of the USA. They cluster closely with but remain distinct from the eastern US green sea turtle cluster. Similarly, the ChHV5 genome from the olive ridley exists in its own group, and is an intermediary between the two larger clusters. The olive ridley individual stranded in Texas (Gulf of Mexico) and so may host a slightly different variant to the ChHV5 genome from the USA east coast. Additionally, the olive ridley individual was partially decomposed upon stranding [55], so the true extent of ChHV5 sequence diversity may not have been recovered due to sample degradation. Samples generally clustered according to geographic location of stranding, with the exception being the four Kemp’s ridley samples sequenced with viral enrichment. However, these four samples may group with the reference genome due to limited reads covering many coding genes (see below), despite a high overall genome coverage.

### 3.3. Phylogenetics Highlights Close Relatedness of Novel Sequences and ChHV5 Florida Variants A–C

As there is only one publicly available comparative whole genome sample (the reference ChHV5 genome), it was not possible to generate a more global ChHV5 genome phylogeny with broader geographic variants. Such analyses will only become possible as whole genome sequencing becomes more widely applied to ChHV5. Therefore, we next used single gene (UL30; *DNA polymerase*) phylogenetics (for which more geographically diverse ChHV5 gene-level sequencing data exists) to explore the geographical relationship between ChHV5 sequenced from our samples and those of previous studies. As, even within UL30 studies, there is a disparity between the number of available sequences depending on fragment length and gene position used, we opted to analyse two sets of UL30 data, one using a longer 2019 bp gene fragment, but with fewer available sequences, and a second using a shorter 483 bp UL30 fragment, for which more sequences from a wider geographic range are available. Only samples generated from this study with over 50 UL30 aligning reads (Appendix A) were selected for UL30 gene analysis. Of the 20 novel samples, all 4 virally enriched Kemp’s ridley samples were excluded due to having insufficient TPM coverage for this gene.

The 2019 bp UL30 gene fragment revealed three major clades, with some smaller sub-groupings. Interestingly, in the largest clade are all 13 samples derived from the novel green turtle samples included in this study (Figure 3A). All of the novel green turtle ChHV5 UL30 sequences, and the two included non-virally enriched novel Kemp’s ridley sequences, clustered closely with the previously reported Florida ChHV5 variants A–C (which are almost identical) [4], although remain distinct. Across the 6801 bp used to define these variants, there is only 9 bp differences between variants A–C, but 383 bp differences in variant D, while there are 145 bp differences between Florida variant A and the Hawaiian variant sequence [4]. This further confirms that the variants infecting green and Kemp’s ridley sea turtles are very similar [23] (Figure 3A). The two Hawaiian ChHV5 UL30 sequences (one reference and one variant) form their own distinct clade, with the two differing only slightly (0.0005 substitutions per site). These phylogenetic results concur with previous findings [4,23,55], highlighting that similar variants of ChHV5 can be present in sympatric species.

Analysis of the shorter ChHV5 UL30 partial gene fragment (~483 bp), from generated sequences, concurs with the previous, larger gene fragment analysis, with all novel ChHV5 samples clustering closely with Florida variants A–C (Figure 3B). This grouping also includes a ChHV5 sequence originating from the Caribbean (Accession: AF299110.1).

Interestingly, all of the Brazilian, Puerto Rican and Gulf of Guinea ChHV5 UL30 sequences (Accession: JN580283.1, JN938586.1, JN938587.1, JN938585.1, JN580280.1, HM348897.1, JN570279.1) form a distinct clade, but more closely relate to Hawaiian ChHV5 in the Pacific rather than the other Atlantic ChHV5 variants, in the context of this small partial gene fragment. When only this short UL30 fragment is considered, a change in the phylogenetic position of olive ridley (Texas) can be seen, with it now grouping within the large clade of the novel green turtle ChHV5 and Florida variants A–C (Figure 3B), whereas before it was separate and somewhat intermediary between the Hawai’i and Florida variants (Figure 3A). This finding also confirms previous analysis of the UL30 short gene fragment of this sample, which found that the olive ridley ChHV5 grouped with Florida variants A–C (Frandsen et al., 2021).

Oddly, a geographically diverse ChHV5 clade containing one olive ridley from the Gulf of Mexico (Accession: AF299109.1, not the olive ridley sequence from this study), one green turtle from the eastern Pacific and one *C. caretta* from Florida (variant D) can be seen. It is possible that the Californian Pacific ChHV5 sample clusters here because it is a relatively shorter UL30 gene sequence (401 bp) than the others, where potential diversity is missing from the 82 bp difference. The two ChHV5 samples from olive ridley are in completely separate clades, further highlighting likely ChHV5 regional evolution and diversity.

### 3.4. Patient “Yucca” Whole Genome Phylogenomics: Do Separate Tumours in the Same Individual Harbour Differing ChHV5 Variants?

We next assessed whether an individual could harbour more than one variant of ChHV5 simultaneously, or if all retrieved ChHV5 sequences from discrete tumours from one individual turtle were identical.

While all ChHV5 sequences obtained from Yucca’s (patient ID: 49-2019-Cm, female; Figure 4B) tumour samples clustered with Florida variants A–C at the UL30 gene level (Figure 3A,B), there were some differences in the ChHV5 sequences between these Yucca tumour samples. Across the full genome, Yucca’s tumour ChHV5 derived genomes varied to the reference genome with a nucleotide diversity range of 1.96% to 2.05% (Figure 4A). When the two most divergent Yucca ChHV5 genomes (samples yuRKTW and yuRKTM) were compared directly with each other, the inter-tumour variance of ChHV5 nucleotide diversity within this individual was 0.15%. There were 198 base-pair differences between the full viral genomes obtained from tumour yuRKTW and tumour yuRKTM. Interestingly, the two samples with the greatest nucleotide variation were both kidney tumours (fibromas). These two samples (yuRKTW and yuRKTM) are from separate tumours, both of which were present in Yucca’s right kidney. yuRKTG was a third tumour from the right kidney, which clustered most closely with yuRKTM (Figure 4A).

## 4. Discussion

### 4.1. Nucleotide Diversity of ChHV5

The severity of wildlife diseases is becoming increasingly exacerbated by anthropogenic activity, a trend projected to worsen over coming decades [62,63,64,65]. The increasing geographic spread of the sea turtle FP global epizootic that is threatening sea turtle conservation is likely driven by rising human-related detrimental changes in the marine environment. Better understanding of FP’s likely aetiological agent, ChHV5, is crucial to determine the contribution of viral evolutionary versus environmental factors in driving this spread, and for mitigation and epidemiological modelling efforts. We report here that the number of SNPs in each of our green turtle tumour derived consensus ChHV5 genomes ranged from 2392 to 2793 when the eastern USA ChHV5 genomes were compared to the Hawaiian ChHV5 reference genome. A total of 1001 fixed nucleotide differences in ChHV5 partial genome sequences were obtained from Pacific (Hawaiian) and Atlantic green turtle populations by a previous study [39]. We also identified between 88 and 2284 SNPs in ChHV5 genomes from Kemp’s ridley and olive ridley turtles compared with those of the Hawaiian (green turtle derived) ChHV5 reference genome. Viral enrichment prior to sequencing resulted in a substantially lower level of SNP detection; however, this is potentially because limited extra-cellular lytic virus is present in FP tumours [20,23,24,25]. Together, these results suggest that significant ChHV5 genetic variation can occur, and that current variant calling based on short stretches of ChHV5 nucleotides likely underrepresents the true extent of ChHV5 variants globally. In the genomes assessed here alone, the maximum sequence divergence was 2.1% of the ChHV5 genome, between the Hawaiian ChHV5 reference genome and that of a ChHV5 genome obtained from a green turtle that stranded in Ormond Beach, northeast Florida, USA.

Thus far, no ChHV5 variants have been tightly correlated to FP disease severity [11,16,39,66,67]. While ChHV5 variants may not determine disease severity, with environmental co-factors and host immune systems potentially playing a larger role [7,11,25], it is also possible that previous studies utilising small cohorts of genes to distinguish variants may have missed some more subtle nucleotide diversity between variants. There is the potential that low level variation (SNPs) within individual genes may correlate with disease severity, a feature not amenable to investigation when only using small fragment sizes for variant calling. For example, slight nucleotide divergence within the human Sars-CoV-2 virus have been linked to increased ease of transmission, enhanced immune evasion, more severe illness and vaccine escape [68]. Next generation sequencing approaches are more amenable to the detection of such variation and should be more widely applied to ChHV5. While unlikely to rival Sars-CoV-2 variant monitoring programmes, the continued cost reduction of NGS approaches [69,70,71] means that they are also likely to become more widespread for non-zoonotic wildlife disease [63,72]. As revealed here and in Morrison et al. (2018) [39], ChHV5 genomes harbour many SNPs, but the potential functional changes of these SNPs remain unknown.

The multi-species ChHV5 whole genomes generated for this study represent an important resource for assessing the geographic sequence diversity of ChHV5, and quantifying changes to the viral genome over time. Future genomic studies across broader geographic areas will be important for revealing the full extent of the existing diversity of ChHV5 worldwide. A study by Patrício et al. (2012) [48], using substitutions per site per year analysis, indicated that ChHV5 is likely under faster evolution than expected for a herpesvirus, although this finding is based on partial sequences of just four ChHV5 genes (DNA polymerase, 483 bp; UL18, 1212 bp; UL34, 861 bp and glycoprotein B, 2486 bp). Patrício et al. (2012) [48] observed an average of 1.32 × 10^-4^ to 4.97 × 10^-4^ substitutions per site per year, on average. Here, we investigated selection pressures exerted on every ChHV5 gene. Given that population demography influences for such a globally widespread pathogen should be minimal, the Tajima’s D scores likely represent the evolutionary pressure each ChHV5 gene is under [39,57,58]. There was a broad range of pressure exerted across the ChHV5 genome, with some genes being constrained by evolution and others under selective pressure to diversify. These results corroborate those reported by Morrison et al. (2018) [39] (across nine tumour samples, including the reference genome (6 Hawai’i/3 Florida)) across a smaller cohort of ChHV5 genes (approximately 40% fewer genes) which also demonstrated a wide range of Tajima’s D values across individual genes. However, for many genes, the Tajima’s D value and direction (positive or negative) varied between the two studies (Appendix A). This emphasises the need to analyse a greater number of ChHV5 genomes, sourced across wider geographic areas, to comprehensively identify diversification between variants. The specific differences in values for some genes likely occur as the Morrison et al. (2018) [39] study contained a greater proportion of ChHV5 genomes from Hawai’i and only studied green sea turtles, whilst the majority of our genomes originated from the eastern US (predominantly Florida), and covering multiple species.

The genes with the highest positive Tajima’s D (balancing selection, higher frequency of maintained allele diversity) were F-UL36 and F-UL37 (0.84 and 1.31, respectively), both of which are essential for viral replication in alphaherpesviruses (Table 2) [73]. Another gene of interest under balancing selection was F-UL41 (0.56, Table 2), a host shutoff protein, which is key to evading the innate immune system. Variance in this gene is likely beneficial to ChHV5, perhaps enabling it to evade innate immunity across diverse populations and species. In contrast, the gene with the lowest Tajima’s D (positive selection, represents excessive low-frequency SNPs) was UL10 (−1.94, Table 2), otherwise known as glycoprotein M (gM) [74]. This gene is highly conserved between alphaherpesviruses, and it participates in multiple phases of the viral life cycle. The majority of ChHV5 genes are practically silenced in FP; however, F-UL10, F-UL36 and F-UL41 were among the small sub-set of ChHV5 genes with active expression in FP tumours, as detected by RNA-seq in a previous study [7]. F-UL10 and F-UL36 expression in particular was detected in multiple tumour tissue types (lung, kidney and external tumours), whereas F-UL41 was only detected in external tumours [7].

The ChHV5 genes identified here with a higher frequency of maintained allele diversity (balancing selection) (Figure 1B) may serve as good candidates for future ChHV5 phylogenetic analyses given the high variability between these genes in different ChHV5 genomes.

### 4.2. ChHV5 Phylogenomics

To avoid the bias and reduced sensitivity that can occur with single-gene phylogenetics, it is recommended that more studies from diverse geographic locations begin to adopt whole viral genome approaches. As has been highlighted by the ongoing human COVID-19 pandemic, viral genome sequencing can be an efficient and rapid means of assessing global viral diversity and of identifying variants of concern, even in an actively evolving situation [68].

Phylogenomic analysis of this study’s ChHV5 genomes revealed clustering based predominantly on geographic location. Some Kemp’s ridley ChHV5 clustered more closely to the Hawaiian reference genome compared to other Florida ChHV5 variants (Figure 2). This contradicts previous evidence, that ChHV5 tends to cluster by geographic location, rather than by species infected [4,23,48,55]. However, only Kemp’s ridley samples virally enriched prior to sequencing clustered with the Hawaiian reference. Despite the high ChHV5 aligning read numbers of these samples, many genes had low read counts, suggesting the majority of reads may have aligned to non-coding regions. Therefore, this proximity to the Hawaiian reference genome may more likely reflect a lack of sufficient reads in coding regions to adequately resolve their true phylogenomic position.

### 4.3. ChHV5 Phylogenetics

This study’s single-gene phylogenetic analyses reaffirm previous findings that ChHV5 has high regional specificity rather than species specificity [4,48,75]. All generated samples were closely related to, but distinct from, Florida variants A–C, with Kemp’s ridley clustering closest to these variants, followed by all green turtle-derived ChHV5 samples (Figure 3A), further highlighting ChHV5 interspecies transmission, despite this being considered abnormal for herpesviruses [76]. Such interspecific transmission suggests that host or environmental exposure differences likely drive the FP prevalence rates observed between species, rather than solely as a function of species-specific ChHV5 variants.

Differences occurred in the phylogeny of some samples between the two versions of the UL30 phylogenetic trees (Figure 3A,B). The larger UL30 sequence tree (2019 bp; Figure 3A) had distinctive clades among the Florida variants A–C, all green turtle and the two Kemp’s ridley samples. However, using a smaller UL30 gene sub-section (483 bp), to allow for a broader geographic comparison, these ChHV5 samples all clustered together in the same clade (Figure 3B). This is likely due to the diversity present in the larger fragments being lost, and highlights a potential issues in determining true phylogeny using small segments of DNA rather than whole genomes. Another clear example of changes in phylogeny placement (loss of resolution), based on the selected sequence, can be seen from the olive ridley sample. This sample was distinct from Florida variants A–C at the whole genome and large UL30 gene fragment level, but groups with the Florida variants A–C using the smaller UL30 gene fragment [55].

### 4.4. Within-Host Viral Diversity

We investigated whether a single individual turtle could host multiple variants of ChHV5 simultaneously. Of patient Yucca’s seven sequenced FP tumour samples, there was notable variance in the ChHV5 consensus genomes generated. These ChHV5 genomes differed from the Hawaiian reference genome by between 1.96% and 2.05% (Figure 4A), with a maximum Yucca inter-tumour ChHV5 diversity of 198 bp (0.15%). This suggests that either differing variants can infect the same individual, or perhaps more likely that nucleotide changes within the ChHV5 genome can arise in viral genomes within a single host individual. Such within-host diversity can be a crucial mechanism in the development of new viral variants [77,78]. All seven consensus ChHV5 genomes generated from Yucca’s samples clustered closely with the other green turtle samples from this study (at both the phylogenomic and UL30 phylogenetic level); however, Yucca’s samples did not form a uniform clade and were interspersed with ChHV5 genomes (and UL30 gene fragments) from other individuals. This suggests that, for future studies, sequencing ChHV5 from only a single tumour on an individual may be insufficient for fine-grained ChHV5 diversity analysis, and may miss intra-individual diversity.

## 5. Conclusions

This study greatly increases the ChHV5 genome-level data available for diversity, evolutionary and phylogenomic comparisons of this sea turtle FP epizootic-associated virus. It also provides evidence across all known ChHV5 protein coding genes (for three sea turtle species), that different genes are under highly variable selective pressures. The study also highlights the underappreciated genetic diversity present across ChHV5 genomes. Finally, this study reveals previously unknown genetic diversity in ChHV5 genomes among different tumours arising concurrently within the same individual.

## Figures and Tables

**Figure 1 animals-11-02489-f001:**
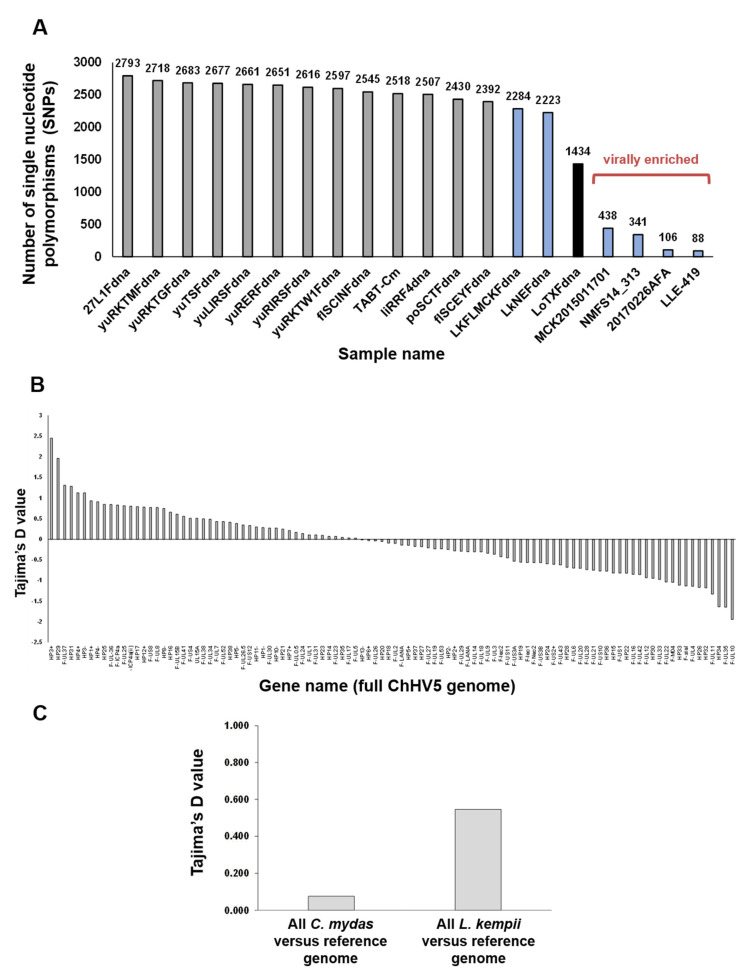
Pan-genome evolutionary dynamics of ChHV5. (**A**) Number of single nucleotide polymorphisms (SNPs) within the ChHV5 genomes from each sample in this study. Colour coding represents turtle origin species of ChHV5 consensus sequence: grey bars represent *C. mydas*, blue bars represent *L. kempii* and black bar represents *L. olivacea*. (**B**) Tajima’s D statistic for every ChHV5 gene (104 genes) generated from all consensus sequences for this study compared with reference ChHV5 genome. (**C**) Tajima’s D for entire ChHV5 genome, all *C. mydas* samples (*n* = 13) pooled versus reference genome, and all *L. kempii* (*n* = 6) samples pooled versus reference genome.

**Figure 2 animals-11-02489-f002:**
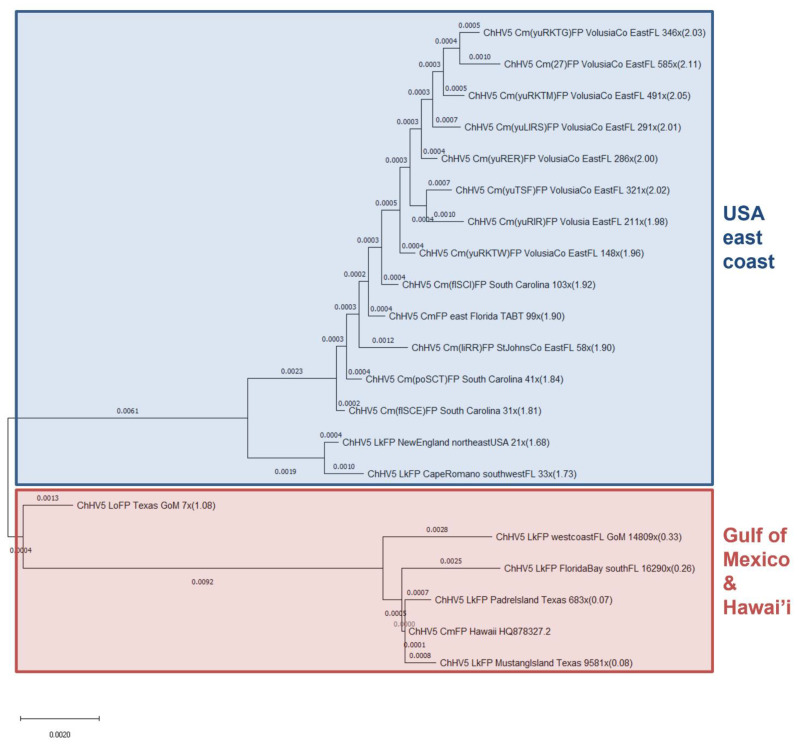
ChHV5 phylogenomic analysis. Whole-genome (132,233 bp) phylogenomic analysis of all ChHV5 genomes, including ChHV5 reference genome (GenBank accession number: HQ878327.2). Analysis of twenty ChHV5 consensus genome sequences generated from 13 individuals. Each ChHV5 genome is listed from left to right by virus name, turtle species (Cm = *C. mydas*, Lk = *L. kempii*, Cc = *C. caretta* and Lo = *L. olivacea*), sample ID, tissue type (FP—tumour), geographic location of turtle, genome coverage and nucleotide diversity from reference genome (%).

**Figure 3 animals-11-02489-f003:**
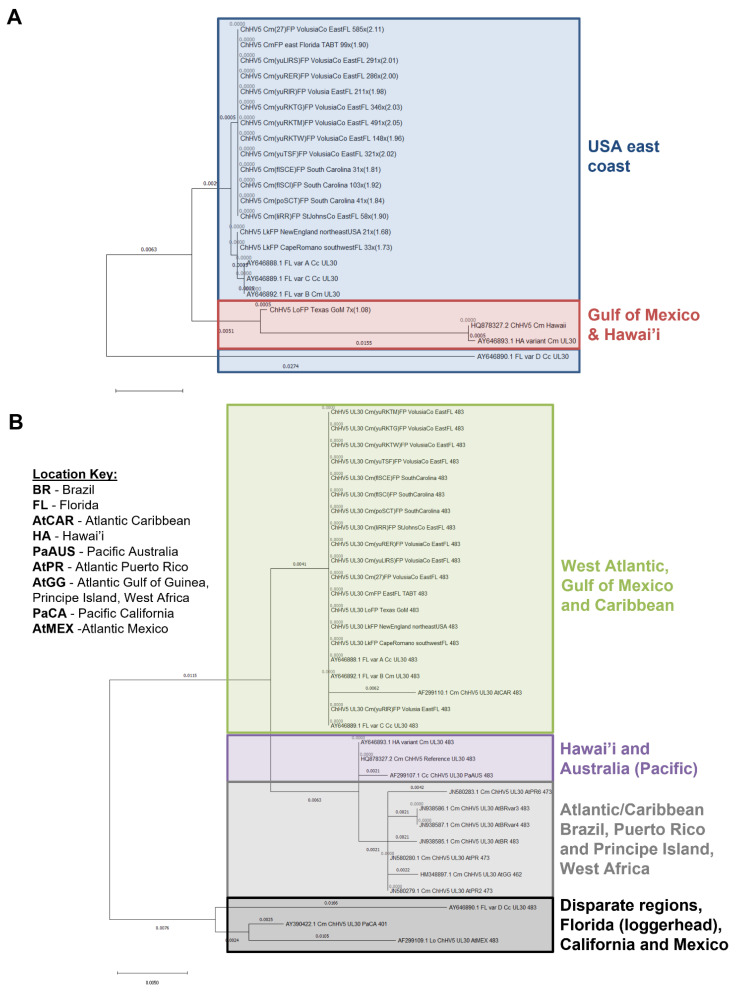
ChHV5 phylogenetic analysis. (**A**) Phylogenetic analysis of partial ChHV5 UL30 gene for generated consensus sequences with sufficient UL30 TPM values, attuned to the same length and position of known full length Florida and Hawai’i variants (variants A–D, HA variant; 2019 bp) from NCBI. All generated sequences are listed by turtle species, tissue type, geographic location of turtle, genome coverage and nucleotide diversity from reference genome (%). (**B**) Phylogenetic analysis of partial ChHV5 UL30 gene for the sixteen generated consensus sequences (with sufficient UL30 TPM values), along with seventeen known sequences from NCBI, all attuned to the same length (~483 bp) and genome position. All NCBI sequences have their unique accession number. All generated sequences are listed by virus, turtle species, tissue type, geographic location of turtle and sequence length (slight length discrepancies between some samples based on deposited sequences).

**Figure 4 animals-11-02489-f004:**
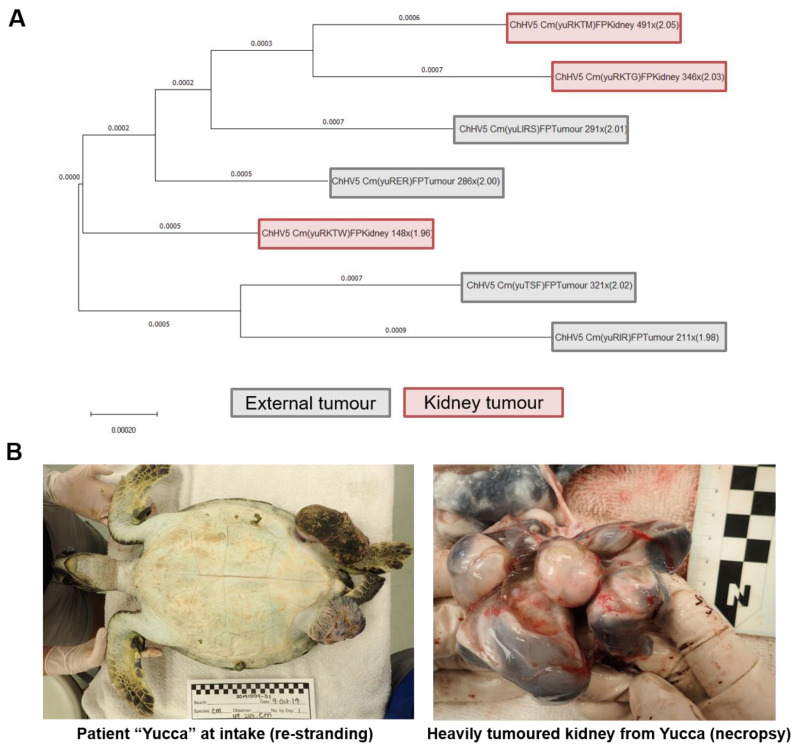
Phylogenomic analysis of ChHV5 genomes isolated from multiple concurrent tumours of a single individual green sea turtle. (**A**) Whole-genome phylogenetic analysis of ChHV5 taken from different FP tumour tissue of one FP-afflicted individual (Yucca, yu, patient ID: 49-2019-Cm). Genome length used for tree generation was 132,233 bp (ChHV5 reference (partial) genome size). In brackets are unique identifiers, followed by FP tissue type (FP Tumour—external tumour, FP Kidney—kidney fibroma tumour), genome coverage and nucleotide diversity as a percentage of the reference ChHV5 genome. Branch figures represent number of substitutions per site. (**B**) Left: Patient Yucca’s hospital intake photo, with large well developed FP tumours visible on her inguinal region. Right: One of Yuca’s heavily tumoured kidneys, imaged during necropsy.

**Table 1 animals-11-02489-t001:** Sequencing information from all samples used for this study. Sample ID, location of stranding, species (Cm: *C. mydas*, Lk: *L. kempii*, Lo: *L. olivacea*), tissue type, sequencing strategy (virally enriched or host and viral), total reads from sequencing, percentage alignment to ChHV5 reference genome, total ChHV5 aligning reads, total ChHV5 reads per 10 million total reads (RPTM) and genome coverage for all samples.

Sample	Stranding Location	Species	Tissue Type	Sequencing Strategy	Total Reads	% ChHV5 Alignment	Total ChHV5 Aligning Reads	ChHV5 RPTM	Genome Coverage
27L1Fdna	Ormond Beach, east Florida	Cm	Lung tumour	No enrichment (host and viral DNA)	6.88 × 10^8^	0.038	257,692.5	3843	584.6×
fISCEYFdna	South Carolina	Cm	External eye tumour	No enrichment (host and viral DNA)	1.31 × 10^8^	0.011	13,627	1064	30.9×
fISCINFdna			External tumour	No enrichment (host and viral DNA)	1.52 × 10^8^	0.03	45,272	3031	102.7×
poSCTFdna	South Carolina	Cm	External tumour	No enrichment (host and viral DNA)	1.52 × 10^8^	0.012	18,133	1224	41.1×
yuLIRSFdna	Halifax Harbour, east Florida	Cm	External tumour	No enrichment (host and viral DNA)	6.21 × 10^8^	0.021	128,132	2119	290.7×
yuRERFdna			External tumour	No enrichment (host and viral DNA)	6.53 × 10^8^	0.02	125,910	1976	285.7×
yuRIRSFdna			External tumour	No enrichment (host and viral DNA)	7.93 × 10^8^	0.012	93,086	1198	211.2×
yuRKTGFdna			Kidney tumour	No enrichment (host and viral DNA)	7.25 × 10^8^	0.021	152,495	2143	346.0×
yuRKTMFdna			Kidney tumour	No enrichment (host and viral DNA)	7.07 × 10^8^	0.031	216,336	3127	490.8×
yuRKTW1Fdna			Kidney tumour	No enrichment (host and viral DNA)	6.30 × 10^8^	0.011	65,453	1063	148.5×
yuTSFdna			External tumour	No enrichment (host and viral DNA)	8.59× 10^8^	0.017	141,446	1682	320.9×
TABT-Cm	Anchorage Marina, east Florida	Cm	Bladder tumour	No enrichment (host and viral DNA)	1.69 × 10^8^	0.03	43,563	2573	98.8×
liRRF4dna	Marineland Beach, east Florida	Cm	External tumour	No enrichment (host and viral DNA)	4.27 × 10^8^	0.01	38,503	391	58.2×
LoTXFdna	Texas	Lo	External tumour (deceased)	No enrichment (host and viral DNA)	1.59 × 10^8^	0.002	3018	192	6.9×
LkNEFdna	New England, Cape Cod, Massachusetts	Lk	External tumour	No enrichment (host and viral DNA)	1.12 × 10^8^	0.01	9062	811	20.6×
LkFLMCKFdna	Cape Romano, SW Florida	Lk	External tumour	No enrichment (host and viral DNA)	1.50 × 10^8^	0.01	14,929	992	33.9×
20170226AFA	Mustang Island, Texas	Lk	External tumour	Viral enrichment	3.45 × 10^7^	14.61	4,222,944	1,461,284	9580.7×
LLE-419	Padre Island, Texas	Lk	External tumour	Viral enrichment	1.46 × 10^7^	2.2	301,155	219,649	683.2×
MCK2015011701	West Coast, Florida	Lk	External tumour	Viral enrichment	5.89 × 10^7^	13.1	6,527,585	1,309,565	14,809.3×
NMFS14_313	Florida Bay, Florida	Lk	External tumour	Viral enrichment	3.69 × 10^7^	23.2	7,180,057	2,320,447	16,289.6×

**Table 2 animals-11-02489-t002:** Ten genes with highest, and 3 genes with lowest Tajima’s D statistic, of genes which have high Transcripts Per Million (TPM) values. Tajima’s D statistic, under the right demography, can infer which genes are likely under positive/reduced selection. Gene predicted features follow Ackermann et al. (2012) [53].

Gene	ChHV5 Genome Position	Tajima’s D Statistic	Predicted Feature
F-UL37	98894–102139	1.31	Tegument protein
F-UL36	91944–98897	0.84	VP1/2 tegument protein
F-UL25	72184–73857	0.81	99% ID with gb|AAU93323.1 minor capsid protein
HP17	57297–57764	0.79	Hypothetical protein (HP)
F-US8	12220–13842	0.77	Glycoprotein e (gE)
F-UL8	43118–45361	0.77	Herpesvirus DNA helicase/primase complex associated protein
HP16	50429–51067	0.66	HP; predicted bipartite NLS
F-UL15B	55718–56788	0.61	Probable DNA packing protein, C-terminus
F-UL41	104719–105897	0.56	Close similarity to gb|AER28066.1. Tegument host shutoff protein
F-US4	14752–15549	0.51	Similar to glycoprotein D (gD)
HP34	126003–126476	−1.64	HP
F-UL35	91561–91926	−1.65	VP26 basic phosphorylated capsid protein
F-UL10	47779–49044	−1.94	Glycoprotein M (gM)

## Data Availability

All raw data and code produced during this study is included in the Appendix A and/or uploaded to Github (code deposited in Github: https://github.com/klyetsko/Whitney-SeaTurtle-FP, accessed on 5 August 2020); sequencing data including raw reads are deposited in NCBI (https://www.ncbi.nlm.nih.gov/, accessed on 5 August 2020) under BioProject ID: PRJNA449022 (https://www.ncbi.nlm.nih.gov/bioproject/PRJNA449022, accessed on 5 August 2020). All ChHV5 consensus sequences used in this study have been deposited into Dryad (https://doi.org/10.5061/dryad.wwpzgmsk6, accessed on 5 August 2020).

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
