# Peer review of "Evolutionary Comparisons of Chelonid Alphaherpesvirus 5 (ChHV5) Genomes from Fibropapillomatosis-Afflicted Green (Chelonia mydas), Olive Ridley (Lepidochelys olivacea) and Kemp’s Ridley (Lepidochelys kempii) Sea Turtles"

_animals, 2021, doi:10.3390/ani11092489_

Round 1
Reviewer 1 Report
The manuscript title: "Evolutionary genomic comparisons of chelonid herpesvirus 5 (ChHV5) from fibropapillomatosis-afflicted green (Chelonia mydas), olive ridley (Lepidochelys olivacea), and Kemp’s Ridley (Lepidochelys kempii) sea turtles" appears very interesting and relevant providing further insights into the pathogenicity of this disease associated to a herpesvirus. However, there is a clear bias in the main conclusions reached in the publications, which contempled only samples from the Atlantic coast of the US excluding all the diversity of a worldwide disease across seven species of marine turtles. Moreover, there are also several flaws in the methodology, writing and final interpretations in this document.
Reviewer 2 Report
A few minor comments might be to include why the 4 Lk samples were treated differently with regards to preparation for analysis. In most cases, my comments are very minor and refer only to slight areas of clarification to help the reader flow through the document, and minor format questions/comments on some of the figures. Finally, the only question, curiosity mostly but would be great to see addressed, is whether the diversity in ChHV5 genomes discovered from one tumor to the next within the same individual could have potentially impacted the clustering/grouping of your analysis with the other individuals from which samples were collected for this study.
Other than these minor points, this is a fascinating and much needed study on this topic! It provides foundational understanding on the true breadth of diversity of this pathogen, its propagation, and potential for manifestation that sets the stage for future work on FP. This manuscript represents key information on which to build our understanding of our role in the spread and diversity of this disease, as well as the full implications for sea turtle recovery and survival in the currently changing environment. Well done!

Reviewer 3 Report
The information provided in the document entitled: Evolutionary genomic comparisons of chelonid herpesvirus 5 (ChHV5) from fibropapillomatosis-afflicted green (Chelonia mydas), olive ridley (Lepidochelys olivacea), and Kemp’s Ridley (Lepidochelys kempii) sea turtles, is very interesting, relevant, has important applications in the area of sea turtle research and conservation. This document provides the most comprehensive picture to-date of whole-genome inter-species ChHV5 diversity and genomic data for future comparisons.
The content of the manuscript is adapted to the requirements of the journal and is developed in a correct way. The document is original and has important contributions to complement the study of sea turtles fibropapillomatosis associated to ChHV5. It has descriptive and scientific support, methodological solidity and describes the results in a simple and direct way.-
The title reflects appropriately the contents of the paper however I suggest to use chelonid “alphaherpesvirus”, since it is the correct name of the etiological agent associated with Fibropapillomatosis. This is indicated in the document.
-
The information provided in the document is very interesting, relevant and can be used as a reference for genomic data comparison.
-
The text is presented in a manner that scientists in other disciplines will understand.
-
The text presented is clearly and concisely.
-
It is a current study with appropriate techniques that represents an advance in the knowledge of sea turtles' diseases.
-
The introduction is complete, informative, clearly demonstrates the details about the study, adequately defines the research problem and presents the background underlying the research. However, I suggest reviewing the comments made in the pdf on lines 61 and 68 of this section.
-
The methodology is complete, clear and understandable, shows the techniques used to be replicable and applicable.
-
The results section is adequate, the results are valid and reliable, graphically demonstrate the findings of the study and highlights the importance of the ChHV5 genomic variants. However I suggest reviewing the comments made in the pdf on lines 449, 258 and 459 of this section.
-
The discussion section is adequate and have been justified sufficiently, highlighting the importance of ChHV5 in different sea turtle species and contrasting with previous information. Additionally, this document provides valuable information for the study of ChHV5 both in the US and in other parts of the world. All relevant aspects of the topic are presented fully. Please review the comments made in the pdf on lines 540 and 589-592 of this section.
-
The literature cited is sufficiently critical, current, and internationally evaluated.
